# Speciation Analysis Method of Heavy Metals in Organic Fertilizers: A Review

**Juan Wang** [1,†]**, Xuejing Wang** [1,2,†]**, Guoxue Li** [2]**, Jingtao Ding** [1]**, Yujun Shen** [1,*]**, Di Liu** [3]**, Hongsheng Cheng** [1]**, Ying Zhang** [1] **and Ran Li** [1]

1  Academy of Agricultural Planning and Engineering, Key Laboratory of Technologies and Models for Cyclic Utilization from Agricultural Resources, Ministry of Agriculture and Rural Affairs, Beijing 100125, China
2  College of Resources and Environmental Sciences, China Agricultural University, Beijing 100125, China
3  Heilongjiang Academy of Agricultural Sciences, Key Laboratory of Combining Farming and Animal Husbandry, Ministry of Agriculture and Rural Affairs, Harbin 150086, China
*  Correspondence: shenyj09b@163.com
†  These authors contributed equally to this work.

**Abstract:** Heavy metals in organic fertilizers pose a risk to the agricultural ecosystem. The environmental risk of heavy metals depends not only on the total amount but also on the speciation. Hence, more information on heavy metals speciation in organic fertilizers is needed to avoid adverse effect. At present, the speciation information of heavy metals is usually obtained by the single-extraction method and sequential extraction method. Common heavy metals that have received attention include Cu, Zn, Pb, Cd, Cr, Hg and As. There is a lack of reviews on speciation analysis methods for heavy metals, specifically in organic fertilizers. This work aims to comprehensively review the methods, explore the problems of the sequence extraction procedure and summarize the factors affecting the distribution of heavy metals speciation. Each sequence extraction procedure of heavy metals in organic fertilizers is described in detail, and the affecting factors are proposed. The review could contribute proposing the directions of optimizing the sequence extraction procedure of heavy metals in organic fertilizers in the future.

**Keywords:** sequence extraction procedure; single-extraction method; speciation; Tessier; BCR; modified Tessier

## 1. Introduction

With the improvement of people's life level, the development of modern agriculture and large-scale farming has become the major trend. As a result, the production of agricultural wastes and livestock and poultry manure has rapidly increased as well. These wastes can be treated by making biomass fuel [1] or composting. In China, the total amount of livestock and poultry manure was stable at 3.7 billion tons in 2010–2020 [2]. To inhibit harmful pathogenic bacteria and to promote growth, metal element additives, including Cu, Zn and As, were often added to feed. But the absorption rate of these elements in livestock and poultry is very low, and more than 95% of heavy metals will be excreted [3]. In the composting process, the organic matter of livestock manure is mineralized and humified, and some of the material is volatilized and lost. So, the concentration of heavy metals often doses be subject to a "relative concentration effect" in composting process, characterized by an increase in the concentration of heavy metals [4]. When this livestock manure is applied to the agricultural soil, the heavy metals will migrate and transform into the soil–plant system and will do harm to human health throughout the food chain [5]. The bioavailability of heavy metals in organic fertilizers is not only dependent on the total amount of heavy metals but also on their speciation. Thus, the method of obtaining concentrations of heavy metals in various speciation is very significant.

It is generally recognized that the harm of heavy metals depends not only on the total amount of heavy metals but also on their specific chemical speciation and binding

states (coprecipitation with minerals, complexation with organic ligands, etc.) [6–8]. The significance of speciation determination is to understand the state of heavy metals in the solid environment and to evaluate the environmental risk according to the different proportions of different speciation. Therefore, obtaining the distribution of various speciation of heavy metals is of great significance for understanding the behavior of heavy metals and environmental risk assessment.

The speciation of heavy metals includes the valence state at the chemical level and the bound state at the operational level [9]. The heavy metal speciation described in this paper refers to the bound state at the operational level. The classical method of speciation analysis is a sequential extraction procedure, whose principle is to use a series of selective reagents to simulate a variety of possible environmental conditions and to dissolve different speciation in turn. The commonly used methods include the Tessier method and the BCR method (the Bureau Communautair de Reference). Tessier's method [10], made in 1979 by a researcher named Tessier, is a five-step extraction that classified heavy metal to five speciation: exchangeable, bound to carbonate, bound to iron-manganese, bound to organic and residual. The five speciation are defined by simulating the combination of heavy metals with different substances in different acid–base and redox environments. Among the five speciation, the bioavailability of heavy metals is getting lower and lower, and the environmental risk is getting smaller and smaller. Because of the shortcomings of the Tessier method, such as poor comparability of results and lack of reference materials for quality control, after discussion by scholars in 1990 [11], the European Community Standards Agency (BCR) proposed a three-step extraction method [12] (BCR method) on the basis of Tessier. Heavy metals were divided into acid extractable state, reducible state and oxidizable state. A sediment reference material CRM601 was developed for the quality control of BCR method.

However, the current speciation analysis methods of heavy metals were developed for soil and sediment and are frequently used for soil and organic fertilizer. The property composition, especially the content of organic matter, is obviously different between organic fertilizer and soil. This will seriously affect the distribution of heavy metals, thus affecting the accuracy of the determination. This paper reviews the sequential extraction methods commonly used in organic fertilizers, the slight changes made by scholars and the differences in property composition and speciation distribution of heavy metals between organic fertilizers and soil. This review will provide a reference for the optimization method of speciation analysis of heavy metals in organic fertilizers.

## 2. Heavy Metals in Manure and Organic Fertilizer

Heavy metals in organic fertilizer from livestock and poultry manure is mainly derived from additives. The common heavy metals include Cu, Zn, Pb, Cd, Cr, Hg and As.

Cu, Zn and Cr are the essential trace elements for animals that participate in metabolism and promote the growth and development of livestock and poultry. Cu has low price and high efficiency. Zn can improve the immunity of the body to prevent diarrhea [13]. Cr strengthens the function of insulin [14]; moreover, Cr can improve the efficiency of feed utilization and save costs. As is an essential trace element for livestock and poultry as well as a toxic element. A right amount of As has a good antibacterial effect, which can significantly enhance disease resistance and speed up metabolism to promote the growth of livestock and poultry. Meanwhile, excessive As can interfere with the normal metabolism of cells, affect the process of respiration and cause cell pathological changes. Pb and Cd are often accompanied in the feed due to different feed additive processes, which are toxic to animals. Excessive levels of Pb and Cd can affect the animals' nervous systems, inhibit hemoglobin synthesis and impair immune function.

It has been shown that the content of heavy metals in feed was significantly and positively correlated with the content of heavy metals in livestock and poultry manure [15]. Therefore, the occurrence of heavy metals in livestock manure and organic fertilizer is inevitable. Cang et al. [16] investigated the heavy metals content of 48 samples of livestock

and poultry manure and 2 samples of organic fertilizer from Jiangsu province of China, in which the livestock species included chicken, pig, cow, duck, goose and pigeon. It showed that the contents of heavy metals in chicken and pig manure were relatively highest, followed by goose and pigeon manure, and the cow and duck manure were relatively lowest. The highest contents of Zn and Cu in livestock manure were up to 77.42–505.9 mg/kg, 14.71–399.0 mg/kg, respectively. Organic fertilizer also had the highest contents of Zn and Cu at 203.37–377.16 mg/kg, 34.16–102.85 mg/kg, respectively [16]. The same pattern was found in 21 samples of organic manure and 133 samples of livestock manure from all over China by Liu Rongle et al. [17] as well, in which the highest content of Zn and Cu in organic manure was 275.8–615 mg/kg, 72.6–414.7 mg/kg, respectively. It is also worth noting that the contents of most heavy metals in manure followed the order: pig manure > chicken manure > Cow manure ≈ sheep manure. Dong Zhanyong et al. [18] examined 20 pig manure samples in Hangzhou China and found that the highest content of Cu and Zn in pig manure was about 300 mg/kg, followed by Cr and As around 6 mg/kg, and the contents of Cd, Pb and Ni were below 1.5 mg/kg.

In summary, pig and chicken manures generally contain more heavy metals than other livestock and poultry manures. In addition, the heavy metal contents in livestock and poultry manures follows the order: Zn > Cu > Cr > Pb > Ni > As ≈ Cd > Hg for most manures. The order of heavy metal contents in organic fertilizer is similar to that in manure.

## 3. Speciation of Heavy Metals in Organic Fertilizers

The total amount of heavy metals in livestock and poultry manure is insufficient to assess their environmental risk because their bioavailability was mainly decided by their speciation in organic fertilizers. There has not been a uniform definition and classification for heavy metals speciation in organic fertilizers so far. It has been pointed out that heavy metals speciation can be classified into valence, chemistry, bound state and structure, according to different perspectives [19]. At present, the commonly used chemical extraction procedure is to classify the heavy metals speciation from the perspective of the bound. According to different experimental purpose and operation processes, the extraction procedures of heavy metals speciation in organic fertilizers are classified into two types: single-stage extraction and sequence extraction procedure (SEP).

The single-stage extraction was to obtain the target heavy metals speciation which was get by only one extractant. Organic fertilizers have similar properties to soil, so extraction methods directly from the soil or with slight modifications were always adopted by researchers [20–23]. Extraction agents were dilute acids; natural or synthetic complexing agents, including dilute hydrochloric acid solution; Calcium chloride solution; Diethylenetriaminepentaacetic acid–Triethanolamine (DTPA–TEA) solution, etc.

The heavy metals speciation was distinguished by successive extraction of a series of extractant, is called for the SEP, which of soil heavy metals have been used to extract various speciation of heavy metals in organic fertilizer as well. Tessier's SEP classified soil heavy metals into five speciation, including exchangeable, bound to carbonates, bound to Fe–Mn oxides, bound to organic matter and residual speciation. The classification was based on different binding modes of heavy metals in soil components [10]. The exchangeable fraction of heavy metals was generally considered to be bioavailability; moreover, the fraction of the carbonate-bound, Fe–Mn oxide-bound and organic-bound were potentially bioavailability, while the residue fraction was hard for absorption by plant. This sequence extraction process mimicked the environmental conditions that may exist in the sediment: the exchangeable extraction process simulated ion exchange conditions; the carbonate-bound extraction process simulated acidic conditions; the Fe–Mn oxide-bound extraction process simulated reducing conditions; the organic-bound extraction process simulated oxidizing conditions; and the residue extraction process was the speciation not easily released into the environment under all conditions.

The "European Community Bureau of Reference Materials" proposed a three-step BCR extraction method [12], which divided heavy metals of soils into three speciation:

acid-extractable; reducible; and oxidizable speciation. Compared with Tessier's method, the BCR method is easier to operate, and speciation is more stable, meanwhile the information obtained on the speciation of heavy metals is far less. To check the efficiency of the extraction process and to make self-examination easy, Ure et al. proposed a four-step BCR method, adding the extraction of residue [24]. In addition, Amacher classified heavy metals speciation into five speciation: exchangeable; bound to carbonate; bound to oxide; bound to organic; and residue [25]. Emmerich classified heavy metals speciation into five speciation: exchangeable; adsorbed; bound to organic; bound to carbonate; and residue speciation [26].

Although all of the above methods were specific to fractionating speciation of soil heavy metals, they were also frequently used in the extraction of heavy metals in organic fertilizers [26–29]. Some researchers have defined speciation of heavy metals speciation, according to the characteristics of organic fertilizers. For example, He et al. classified heavy metals into five speciation: water-soluble; potassium chloride extractable; organic complexed; bound to organic; and mineral, according to the change of organic matter during composting. They referred heavy metals as humic acid bound and fulvic acid bound, respectively, which were extracted by sodium pyrophosphate and sodium hydroxide, according to the change of humus [30].

## 4. Extraction Procedure of Heavy Metals in Organic Fertilizers

### 4.1. Single-Extraction Method of Heavy Metals Speciation in Organic Fertilizers

The single-stage extraction method dissolved a specific speciation directly by an extractant only and was usually used for extracting some specific speciation, such as the active, migrated, bioavailable and plant-available speciation. It is a simple, time-saving and effective way to determine the level of heavy metals contamination in organic fertilizers visually. There were a wide variety of extraction agents, including deionized water, acid, neutral salt, chelating agent [31]. The common extractants and operating conditions of single-stage extraction are shown in Table 1.

(1) Migration speciation [32,33] Migration speciation or the soluble complexed speciation presents one of the most migratory speciation of heavy metal. The method used for extract is adding deionized water to the sample and shaking for 16 h.

(2) Acid leachable speciation [34] Acid leachable speciation presents almost the whole fraction of elements that bind to materials by simply absorption. The method used for extract is adding 10 mL 0.5 mol/L HCl into solution. The dry sample is 1g and shaking time is 16 h.

(3) Effective speciation [35] Effective speciation presents one of the most migratory speciation of heavy metal and is the most easily absorbed by plants. The extract agent is Diethylenetriaminepentaacetic acid (DTPA). DTPA is a chelating agent which can form water-soluble complexes with metal ions. For example, the Cu and Zn content in DTPA's extracts were relevant with the heavy metals in plant roots.

(4) Plant available speciation [36,37] Plant available speciation presents the fraction of heavy metals that can be absorbed by plant roots. The method used for extract is adding 0.05 mol/L Ethylenediaminetetraacetic acid (EDTA) to the sample and shaking for 1 h or adding mixed acid, which simulates plant root exudates, to the sample and shaking for 16 h. EDTA could extract heavy metals by dissolution, complexation and adsorption effect. It could form stable water-soluble complexes with heavy metals and is less aggressive in silicates compared with hydrochloric acid. Acetic acid, lactic acid, malic acid and formic acid, which are low molecular organic acids, were dominate in plant root exudates. The application of this acid mixture could mimic the environment in which heavy metals are absorbed by plants and extracted by acidification, chelation and redox reactions. The extraction method using this acid mixture is called rhizosphere-based extraction (REM) and is more realistic than the other method of extracting plant-available speciation of heavy metals.

(5) Bound to humic acid speciation [38] Bound to humic acid speciation presents the complexes of heavy metals with humic acid. Humic acid, which is soluble in alkaline

conditions only, could form water-soluble sodium humate with sodium pyrophosphate. Thus, the methods used for extract is adding 0.1 mol/L $Na_4P_2O_7$ and 0.1 mol/L NaOH to sample and shaking for 24 h.

(6) Leachable speciation [37] Leachable speciation is used to evaluate the leachability of heavy metals. The leachable process of acetic acid extraction was used to simulate the leaching of metals and their coagulation process in sludge samples, also known as toxicity characteristic leaching procedure (TCLP).

(7) Bioavailable speciation [37] Bioavailable speciation is the fraction of heavy metals that can be absorbed by animals. The extraction is adding 0.4 mol/L glycine (pH adjusted to 1.5 by HCl) and shaking for 1 h. Glycine (hydrochloric acid adjusted pH = 1.5) could be made into a synthetic gastric solution for the extraction of bioaccessible heavy metals. This method is also known as the simplified bioaccessibility extraction test (SEBT). In addition, some studies have used other extractants [27]. The bioavailable part of heavy metals could be extracted to the maximum extent by extractant simulating saliva and gastric juice. The composition of these two solutions is as follows: The first solution consists of 4.0 g mucin, 1.0 g urea, 0.6 g $Na_2HPO_4$, 0.6 g $CaCl_2$, 0.4 g KCl, 0.4 g NaCl and 1000 mL deionized water to simulate saliva (pH 5.5), and the second solution (simulated gastric juice, pH 1.5) consists of 1000 mL deionized water, 2 g NaCl, 7 mL concentrated HCl and 3.2 g pepsin. The extraction process was carried out by adding solution one followed by solution two by 2 h. The extraction of Pb by this method showed a high correlation with the Pb extracted by $NH_2OH$-HCl, which was the main extractant of amorphous oxide-bound Pb. Therefore, the oxide-bound Pb was the main speciation extracted by the two simulated biological fluids.

**Table 1.** Single extraction method for heavy metals.

| Speciation | Extractant | Object | Extraction Time | Reference |
|---|---|---|---|---|
| Migration | Deionized water | Soil | 24 h | [32,33] |
| Acid leachable | 0.5 mol/L hydrochloric acid | Sediment | 16 h | [34] |
| Effective | DTPA | Sludge composting | - | [35] |
| Plant available | 0.05 mol/L Ethylenediaminetetraacetic acid (EDTA) | Soil | 1 h | [36] |
| | 10 mmol/L $CH_3COOH$, lactic acid, citric acid, malic acid, and formic acid mixture | Sludge composting | 16 h | [37] |
| Bound to Humic acid | 0.1 mol/L $Na_4P_2O_7$ + 0.1 mol/L NaOH | Composting | 24 h | [38] |
| Leachable | 0.1 mol/L $CH_3COOH$ (pH = 4.93) | Sludge composting | 18 h | [37] |
| Bioavailable | 0.4 mol/L glycine (pH adjusted to 1.5 by HCl) | Sludge composting | 1 h | |

It can be seen that the speciation of heavy metals extracting by various single extraction processes is not fixed. The extracted single speciation includes the most migratory bound, humic acid-bound, plant-available and bioavailable bound. There speciation had strong biological toxic effect except for the humic acid-bound.

### 4.2. SEP of Heavy Metals Speciations in Organic Fertilizers

The Sequence extraction procedure is a method to obtain heavy metals speciation information under different extraction conditions, according to the difficulty level from weak to strong in the extraction process by different extractant [10]. This method is the mainstream method for heavy metals speciation analysis in soil, sediment and organic fertilizer at this stage. More comprehensive information on the speciation of heavy metals can be obtained by this method.

Tessier's method [10] was proposed by Tessier in 1979 and is the most widely accepted method for the extraction of heavy metals speciation. The method was described as follows: the quantities indicated below refer to 1-g samples (dry weight of the original sample used for the initial extraction), ① Exchangeable. 8 mL, 1 mol/L $MgCl_2$ (pH = 7) was added and shaken for 1 h at 25 °C; ② Bound to carbonate. 8 mL, 1 mol/L $CH_3COONa$ was shaken for 5 h at pH = 5 ($CH_3COOH$ adjustment) and temperature of 25 °C; ③ Bound to Fe-Mn oxide. 20 mL 0.04 mol/L $NH_2OH$-HCl (dissolved in 25% $CH_3COOH$) was intermittently shaken at 95 °C for 6 h. ④ Bound to Organic. Extraction was performed in three steps, first with 3 mL of 0.02 mol/L $HNO_3$ and 5 mL of 30% $H_2O_2$ at pH = 2 (adjusted by $HNO_3$) and temperature of 85 °C for 2 h intermittent shaking; then with 3 mL of 30% $H_2O_2$ at pH = 2 (adjusted by $HNO_3$) and temperature of 85 °C for 3 h intermittent shaking; after cooling, finally with 5 mL of 3.2 mol/L $CH_3COONH_4$ (dissolved in 20% $HNO_3$), diluted to 20 mL and shaken for 0.5 h at 25 °C; ⑤ Residue. HF-$HClO_4$ was digested until clarified and clear (Table 2). After each extraction step, the supernatant by centrifugation was stored for measurement, and the precipitate was washed with 8 mL of deionized water before proceeding to the next step. $MgCl_2$, $CH_3COONa$ and $CH_3COONH_4$ released the metal elements mainly by ion exchange, $NH_2OH$–HCl released the heavy metals adsorbed on the surface and co-precipitated with iron and manganese oxides by reducing action. While $H_2O_2$, $HNO_3$, HF and $HClO_4$ released metal by destroying the soil matrix [31]. The Tessier procedure provided an important idea for the extraction of heavy metals speciation and a template for subsequent scholars to improve the method.

However, the comparable results of the Tessier method were poor, and the standard materials for quality control were absent [32]. Thus, the Bureau Communautair de Reference (BCR) [24] proposed a three-step extraction method based on Tessier, which divided heavy metals into acid-extractable, reducible and oxidizable. The acid-extractable speciation including exchangeable speciation and carbonate-bindings [33].

To further enhance the accuracy of the BCR extraction process, A. Sahuquillo et al. [12] proposed an improved BCR method by changing the $NH_2OH$–HCl concentration from 0.10 M to 0.50 M and reducing the $NH_2OH$–HCl solution pH from 2 to 1.5. Meanwhile, the pH adjustment reagent was recommended to use $HNO_3$ instead of HCl because chlorine could speciation dissolved complexes with heavy metals. The relative standard deviation of the results was reduced and the accuracy was greatly improved. This was because heavy metals are easier to be soluble at pH 1.5, and the low pH (1.5) improved the buffering capacity of the extractant (the pH of the solution varied before and after extraction). Regarding the solid-liquid separation method, the filter paper was added into a centrifuge tube in the next extraction step to avoid the loss of solid in the filtration process. However, it increased the concentration of the extract in step 3 significantly. So, filtration was recommended to separate the solid and liquid phases, while the centrifugation speed was increased from $1500\times g$ to $3000\times g$. This method changed Tessier's five-extraction procedure to a three-step extraction procedure that was easier and simpler to handle than the Tessier method.

In view of the rich organic matter in black soil, a 7-step modified Tessier method was studied [39]. The method divided heavy metals into six speciation: water-soluble; bound to carbon; bound to extractable humus; bound to iron-manganese oxide; bound to non-extractable humus; and residue state. The improvement was based on the different characteristics of fulvic acid, huminic acid and hominin. The fulvic acid and huminic acid that can be soluble in alkali were called extractable humic, and the non-extractable humic were referred to as non-alkali soluble hominin. The extractant was applied to 0.5 mol/L NaOH and leached three times, resulting in good recovery and an effective method.

There are a large number of studies on the extraction of heavy metal forms in soils and sediments by SEPs; however, only a few studies on the extraction of heavy metal forms in organic fertilizers can be found. In Figure 1, 28 reports, which studied heavy metals speciation by SEPs in organic fertilizers, are shown.

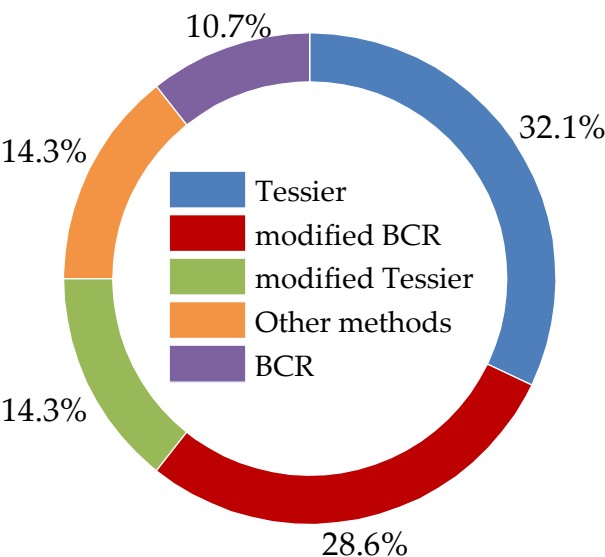

**Figure 1.** Sequential extraction method of heavy metals in organic fertilizers.

It showed that the Tessier method was the most frequently used method with 32.14% of the total articles [21,22,40–47], followed by the modified BCR method with 28.57% [20,23,30,47–51]. The modified Tessier method [52–55] and other methods both accounted for 14.29%. The original BCR method [40,56,57] was the least often used, accounting for only 10.71%. Other methods include Sposito [58,59], Emmerich [26], etc. The procedures of these methods were shown in Table 2.

**Table 2.** Operation steps of continuous extraction method (SEP).

| Speciation | Extractant | Operation Method | Centrifugal and Water Washing Conditions |
|---|---|---|---|
| Tessier method [10] | | | |
| Exchangeable | 1 mol/L MgCl$_2$ (pH = 7) | Solid-liquid ratio * 1:8; shaken at room temperature for 1 h | |
| Bound to carbonate | 1 mol/L NaOAc (pH = 5) | Solid-liquid ratio 1:8; shaken at room temperature for 5 h | |
| Bound to iron-manganese oxide | 0.04 mol/L NH$_2$OH-HCl (25% CH$_3$COOH) | Solid-liquid ratio 1:20; 96 °C water bath with intermittent shaken for 6 h | Centrifuged at 10,000 rpm for 30 min, washed with 8 mL deionized water, and then centrifuged |
| Bound to organic | 0.02 mol/L HNO$_3$, 30% H$_2$O$_2$ (pH = 2), 3.2 mol/L NH$_4$OAc (20% HNO$_3$) | ① Solid-liquid ratio of 1:3 of HNO$_3$ and 1:5 of 30% H$_2$O$_2$, 85 °C water bath for 2 h; ② Enforce liquid ratio 1:3 of H$_2$O$_2$ 85 °C water bath for 3 h until nearly dry. ③ After cooling, strengthen the liquid ratio of 1:5 CH$_3$COONH$_4$ and diluted to 20 mL, shaken at room temperature for 0.5 h | |
| Residual | HF-HClO$_4$ | ① HF-HClO in a solid-liquid ratio of 1:10:2, steamed to near dryness. ② Reinforced the liquid ratio 1:10:1 of HF-HClO$_4$ and steamed until nearly dry. ③ Then added 1:1 HClO$_4$ and steamed until white smoke, 12 mol/L of HCl dissolved, diluted to 25 mL | |

**Table 2.** *Cont.*

| Speciation | Extractant | Operation Method | Centrifugal and Water Washing Conditions |
|---|---|---|---|
| Modified Tessier 1 [54] | | | |
| Exchangeable | 1 mol/L MgCl$_2$ (pH = 7) | Solid-liquid ratio 1:10; 25 °C, 220 rpm shaken for 1 h | |
| Acid-extractable | 1 mol/L NaOAc (pH = 5) | Solid-liquid ratio 1:10; 25 °C, 220 rpm shaken for 5 h | |
| Reducible | 0.1 mol/L NH$_2$OH-HCl (25% CH$_3$COOH) | Solid-liquid ratio 1:10; 96 °C water bath with intermittent shaken for 6 h | Centrifuged at 8000× $g$ and 0.45 μm membrane filtration after 5 min, 10 mL of deionized water washed and centrifuged |
| Oxidizable | 30% H$_2$O$_2$ (pH = 2), 3 mol/L CH$_3$COONH$_4$ (20% HNO$_3$) | ① Solid-liquid ratio of 1:2 in H$_2$O$_2$, digested at room temperature for 30 min, ② mixture was heated to 85 °C for 5 h, ③ After cooling, strengthen the liquid ratio 1:0.8 of CH$_3$COONH$_4$, 25 °C, 220 rpm, and shaken for 0.5 h. | |
| Residual | HF-HClO$_4$ | Solid-liquid ratio 1:10:2 (HF: HClO$_4$), 2 h digestion at 120 °C | |
| Modified Tessier 2 [52] | | | |
| Exchangeable | 1 mol/L MgCl$_2$ (pH = 7) | Solid-liquid ratio 1:10; 20 °C, 200 rpm shaken for 1 h | |
| Bound to carbonate | 1 mol/L NaOAc | Solid-liquid ratio 1:10; 20 °C, 200 rpm shaken for 5 h | |
| Bound to iron-manganese oxide | 0.1 mol/L NH$_2$OH-HCl (25% CH$_3$COOH, pH = 4) | Solid-liquid ratio 1:10; 90 °C water bath, intermittent shaken 6 h | Centrifuged at 8000 rpm for 15 min, 0.45 μm filter membrane filtration, 10 mL of deionized water washed and centrifuged |
| Bound to organic and sulfur | 30% H$_2$O$_2$ (pH = 2) | Solid-liquid ratio 1:10, 90 °C water bath, intermittent shaken for 1 h | |
| Residual | HNO$_3$, 70% HClO$_4$ | 6 mL concentrated HNO$_3$ and 70% HClO$_4$ digestion | |
| Modified Tessier 3 [55] | | | |
| Exchangeable | 0.5 mol/L MgCl$_2$ (pH = 7) | Solid-liquid ratio 1:10; shaken at 25 °C for 2 h | |
| Bound to carbonate | 0.5 mol/L NaOAc-0.5 mol/L CH$_3$COOH, (pH = 4.74) | Solid-liquid ratio 1:10; shaken at 25 °C for 3 h | - |
| Bound to iron-manganese oxide | 0.175 mol/L (NH$_4$)$_2$C$_2$O$_4$ -0.1 mol/LH$_2$C$_2$O$_4$ (pH3.25) | Solid-liquid ratio 1:10; shaken at 25 °C for 3 h | |
| Bound to organic | 30% H$_2$O$_2$, 0.5 mol/L NaOAc-0.5 mol/L CH$_3$COOH (pH 4.74) | ① Solid-liquid ratio 1:2.5 of H$_2$O$_2$; evaporated to near dryness in a water bath at 85 °C and repeated once; ② After cooling, reinforced the NaOAc-CH$_3$COOH buffer solution with a ratio of 1:10 and shaken at 25 °C for 3 h. | |
| Residual | | Add concentrated HNO$_3$ heat on a hotplate until nearly dry, added HClO$_4$ heat until white, and dissolved with 0.1 mol/L HNO$_3$ | |

| Speciation | Extractant | Operation Method | Centrifugal and Water Washing Conditions |
|---|---|---|---|
| **BCR method [12]** | | | |
| Exchangeable/acid extractable | 0.11 mol/L $CH_3COOH$ | Solid-liquid ratio 1:40, 20 °C, 30 rpm shaken for 16 h | Centrifuged at $1500 \times g$ for 10 min; 20 mL deionized water shaken for 15 min and centrifuged |
| Reducible | 0.1 mol/L $NH_2OH$-HCl (pH = 2) | Solid-liquid ratio 1:40, 20 °C, 30 rpm shaken for 16 h | |
| Oxidizable | 8.8 mol/L $H_2O_2$; 1 mol/L $CH_3COONH_4$ | ① Solid-liquid ratio of 1:10 $H_2O_2$, 85 °C water bath until nearly dry, repeated once. ② Added 1:50 of $CH_3COONH_4$ and shaken for 16h at 20 °Cand 30 rpm. | |
| **Modified BCR 1 [51]** | | | |
| Exchangeable | 0.11 mol/L $CH_3COOH$ | Solid-liquid ratio 1:40, 22 °C shaken 16 h | Centrifuged at 3000 rpm for 20 min followed by filtration through a 0.45 μm membrane. |
| Reducible | 0.5 mol/L $NH_2OH$-HCl ($HNO_3$ adjust pH 1.5) | Solid-liquid ratio 1:40, 22 °C shaken 16 h | |
| Oxidizable | 30% $H_2O_2$, 1 mol/L $CH_3COONH_4$ (pH = 2) | ① Solid-liquid ratio 1:40 of $H_2O_2$, 85 °C water bath for 2 h with intermittent shaken; ② Added 1:50 of $CH_3COONH_4$ and shaken at 22 °C for 16 h | |
| Residual | $HNO_3$, HF | Solid-liquid ratio 1:5:1 (HNO, HF) | |
| **Modified BCR 2 [60]** | | | |
| Exchangeable/acid extractable | 0.11 mol/L $CH_3COOH$ | Solid-liquid ratio 1:40, 22 °C, 30 rpm shaken for 16 h | Centrifuged at 3000 rpm for 20 min; 20 mL deionized water shaken for 15 min and centrifuged |
| Reducible | 0.5 mol/L $NH_2OH$-HCl (2.5%, 2 mol/L $HNO_3$) | Solid-liquid ratio 1:40, 22 °C, 30 rpm shaken for 16 h | |
| Oxidizable | 8.8 mol/L $H_2O_2$ (pH = 2), 2 mol/L $CH_3COONH_4$ (20% $HNO_3$) | ① Solid-liquid ratio 1:10 of $H_2O_2$, digested at room temperature for 1 h; followed by a water bath at 85 °C with intermittent shaken for 1 h until 3 mL remained. ② Added another 1:10 of $H_2O_2$ water bath at 85 °C shaken intermittently for 1 h until about 1 mL is left. ③ After cooling, reinforced liquid ratio of 1:50 $CH_3COONH_4$, 22 °C, 30 rpm shaken for 16 h | |
| Residual | HCl (37%), $HNO_3$ (70%) | Solid-liquid ratio 1:7:2.3 (HCl, $HNO_3$) | |

**Table 2.** *Cont.*

| Speciation | Extractant | Operation Method | Centrifugal and Water Washing Conditions |
|---|---|---|---|
| Modified BCR 3 [28] | | | |
| Exchangeable | 0.11 mol/L $CH_3COOH$ | Solid-liquid ratio 1:40, 20 °C shaken 16 h | |
| Reducible | 0.5 mol/L $NH_2OH$-HCl ($HNO_3$ adjust pH 1.5) | Solid-liquid ratio 1:40, 20 °C shaken 16 h | |
| Oxidizable | 30% $H_2O_2$, 1 mol/L $CH_3COONH_4$ (pH = 2) | ① $H_2O_2$ in a solid-liquid ratio of 1:10, reacted at room temperature for 1 h with intermittent shaken; ② 1 h in a water bath at 85 °C with intermittent shaken until the volume is reduced to 1–2 mL; ③ Added 1:10 of $H_2O_2$ again and repeated step ②. ④ After cooling, added 1:50 of $CH_3COONH_4$ and shaken at 20 °C for 16 h | Centrifuged at 4000 rpm for 20 min; washed with 10 mL deionized water for 15 min |
| Residual | $HNO_3$, HCl, HF | ① Solid-liquid ratio 1:20 of $HNO_3$, covered with an electric hot plate at 60 °C for pre-dissolution overnight; ② After cooling, added 1:10:5 HCl, and HF, covered the oven, and heated at 160 °C for 8 h | |
| Sposito [58] | | | |
| Exchangeable | 0.5 mol/L $KNO_3$ | Shaken for 16 h | |
| Water soluble | $H_2O$ | Shaken for 2 h and repeated three times | - |
| Bound to easily migratory organic | 0.5 mol/L NaOH | Shaken for 16 h | |
| Bound to complexed organic or carbonate | 0.05 mol/L EDTA | Shaken for 6 h | |
| Sulfide | 4 mol/L $HNO_3$ | 80 °C water bath for 16 h | |

* Solid-liquid ratio in (g/mL).

Exchangeable speciation is the combination of heavy metal and particulate matter through electrostatic adsorption. Bound to carbonate speciation refers to heavy metals binding with carbonate minerals, which is easily destroyed in acidic environment under the influence of pH values. Bound to Iron-manganese oxide speciation refers to the binding heavy metals with clay minerals such as iron-manganese oxides, which is easy to be destroyed under reduction conditions. Bound to organic speciation refers to the combination of heavy metals and organic matter, which is easy to be destroyed under oxidation conditions. Residual state refers to the remaining forms of heavy metals after the removal of the above four forms, which generally exist in the mineral lattice.

There were many changes to this SEPs method during optimization, which commonly included reducing the solid–liquid ratio, changing the extractant type, extractant concentration, pH, extraction time, centrifugation time, centrifugation speed, solid cleaning procedure and so on. These also are the most sensitive factors in the sequential extraction process. The details of improvement to the Tessier method included the following: decreased the solid-liquid ratio from 1:8 to 1:10; decreased the exchangeable extractant $MgCl_2$ from 1 mol/L to 0.5 mol/L while changing the extraction time from 1 h to 2 h; shortened the carbonate-binding shaking time from 5 h to 3 h; increased the iron-manganese oxide binding extractant $NH_2OH$–HCl from 0.04 mol/L to 0.1 mol/L or changed the extractant $NH_2OH$–HCl to $(NH_4)_2C_2O_4$–$H_2C_2O_4$ solution while decreasing the temperature from

95 °C to 25 °C; changed the organic bound extractant from $CH_3COONH_4$ to $CH_3COONa$-$CH_3COOH$ buffer solution or simplified the organic bound extraction process; changed the $HF$-$HClO_4$ to $HNO_3$-$HClO_4$ in the residue extraction procedure. The main improvements to the BCR method were increasing the concentration of hydroxylamine hydrochloride from 0.1 mol/L to 0.5 mol/L; decreasing the pH of the solution from 2 to 1.5; and strengthening the extraction degree of the residue.

Decreasing the solid–liquid ratio may be for the reason that organic fertilizers are more likely adsorbed to the wall of the tube compared to soil, so increasing the amount of liquid can reduce inadequate solid–liquid contact. Weakening the extraction conditions of the bound to carbonate and Fe–Mn oxide states was a result of the low mineral content in the organic fertilizer as well as to prevent interference in the next extraction step. Increasing the concentration of hydroxylamine hydrochloride while decreasing the pH value could improve the extraction accuracy of the reducible state.

In addition, centrifugation speed and time differed slightly among these methods, such as 30 min at 5000 rpm, 15 min at 8000 rpm, 20 min at 4000 rpm and so on. A short centrifugation time could lead to incomplete solid-liquid separation and a loss of solid particles, which was not encouraged. The centrifugation should be done immediately after the extraction to decreased reabsorption and redistribution [61]. Furthermore, excessive washing time may also lead to a partial loss of heavy metals.

## 5. Factors Affecting the Speciation of Heavy Metals in Organic Fertilizers

The factors influencing the distribution of heavy metals speciation included two types: the physical composition and chemical properties of organic fertilizers and the type of heavy metals.

### 5.1. Property of Organic Fertilizer Affecting Heavy Metals Speciation Distribution

Organic matter is an important factor affecting the distribution of heavy metals speciation. Organic matter comes in various speciation, such as dissolved organic matter (DOM), granular organic matter (POM), humus, etc. The surface of organic matter contained abundant functional groups with which heavy metals can react by adsorption and complexation to organic speciation bound. Soluble organic matter will enhance the mobility of heavy metals, while non-soluble organic matter will reduce the mobility of heavy metals. Meanwhile, the accumulation of organic matter during the formation of organic fertilizer can promote the conversion of heavy metals from the exchangeable to the organic-bound speciation [62], which reduces the biological effects of heavy metals. At the same time, these strong binding processes lead to a significant increase of heavy metals in the organic binding state when the content of organic matter is high. Also, it may affect the integrity of the organic binding state extraction.

It has also been shown that the proportion of heavy metals bound to organic matter increased with the increasing of soil organic matter [63]. Humic substances in organic fertilizers include humic acid (HA), fulvic acid (FA) and humic black matter. HA and FA have a large number of surface functional groups, such as carboxyl, phenolic hydroxyl, alcohol hydroxyl, amides and aldehydes. They involve high-energy bonds [64] and can bind to heavy metals to speciation binary or multiple complexes through acidification, complexation, precipitation and redox reactions, affecting the solubility and stability by reducing the mobility of heavy metals. Zhang et al. [63] classified organic heavy metals as POM and fine particle organic matter in soil humus. POM was a type of organic carbon pool with low humification but high activity. It adsorbed heavy metals with a fast reaction rate, with an equilibrium time less than 100 min. The sorption ability of POM to heavy metals was significantly higher than minerals with the same particle size, and closer or slightly higher to fine soil particles. Fine particle soil humus included soluble and insoluble humus. Insoluble humus could adsorb heavy metals by ion exchange effect, and soluble humus could speciation stable complexes with heavy metals [65]. Therefore, when the content of organic matter is high, both soluble and insoluble organic matter will increase,

leading to the conversion of heavy metals from an exchangeable to organically bound state, and the organic bound state may have strong or weak migration. Therefore, there are certain challenges for the accuracy analysis of exchangeable and organically bound states.

Organic matter came from the decay of livestock manure mixed with straw and wood chips, so the content of organic matter was high. Although the inorganic mineral fraction was relatively low in soil, heavy metals were still present in inorganic minerals. Minerals can combine heavy metals by adsorption, co-precipitation, as well as bonding with high covalent properties, such as iron-manganese oxides, hydroxides, etc. [66]. In addition, a small amount of heavy metals were present in the lattice of silicate minerals.

The pH of organic fertilizers can also affect heavy metals speciation distribution [67,68]. Under acidic conditions, the bioavailability of heavy metals was affected by the sorption process, while under neutral or alkaline conditions, the bioavailability of heavy metals was affected by dissolution, precipitation and complexation [66,69]. When the pH value was low, a higher proportion of the adsorbed heavy metals were exchangeable by dilute salts [69]. Acidic environments favored the release of Fe/Mn by oxides through ligand-promoted solubilization, and they affect the formation of heavy metal complexes, chelates and coordination polymers. Under high pH conditions, heavy metals existed as precipitates in a large proportion. Therefore, under the condition of high pH values, the heavy metal will transform from the exchangeable state with high activity to the iron and manganese oxide binding state or organic binding state with low activity. In order to avoid the influence on the composition of the sample during the extraction of these forms, it is necessary to pay attention to the preparation of acid condition, and pH values of extractant should not be too high.

Organic matter, mineral type and pH conditions were different in soils and organic fertilizers (Figure 2). The organic matter content in soil was relatively stable with a percentage of below 10%, while the organic matter content in organic fertilizer was about 60%, much higher than that in soil; the minerals in soil are 70–95% [70], while the minerals in organic fertilizer are much smaller than soil as its high organic matter; the range of pH value in soil was wider with an average of 5.5, while the range of pH value in organic fertilizer was narrower with an average of 7.5 [69,71–77]. The current methods for extraction of heavy metals speciation were developed based on soil and sediment property. Whether these methods were applicable was debatable, so there was a possibility of bias between the results and the actual values of heavy metals speciation in organic fertilizers.

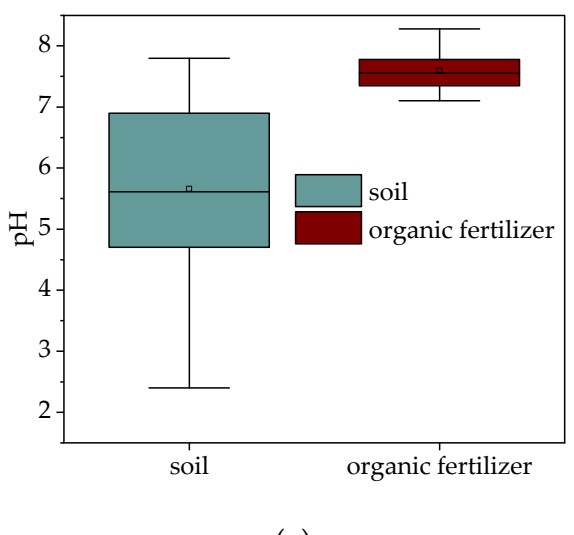

**(a)**

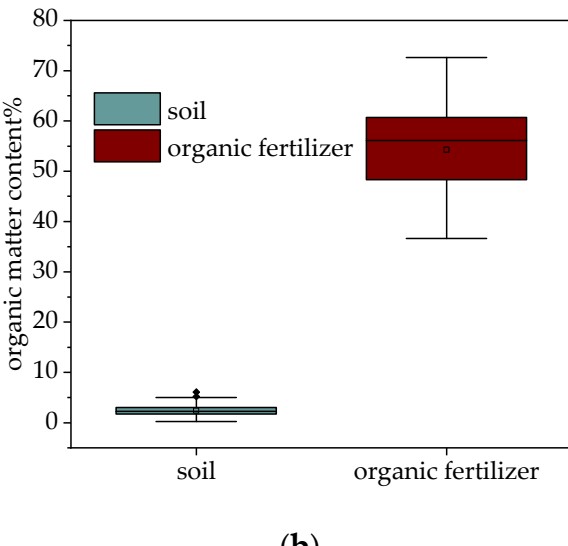

**(b)**

**Figure 2.** The pH value (**a**) and organic matter content (**b**) in soil and organic fertilizer.

### 5.2. Comparison of Heavy Metals Speciation between Organic Fertilizers and Soils

Comparing the total amount of heavy metals in soil and organic fertilizer (Figure 3), it can be found that the content of heavy metals, especially Cu and Zn, in organic fertilizer was much higher than that in soil. The total amount of Zn in organic fertilizer was about 30–550 times (maximum and minimum values) higher than that in soil. The total amount of Cu in organic fertilizer was about 5.8–192 times higher than that in soil. The contents of Cd, Cr, Pb and Ni were about 13.38, 11.99, 8.31 and 20.53 times higher than that in soil, respectively. As and Hg content were close between organic fertilizer and soil.

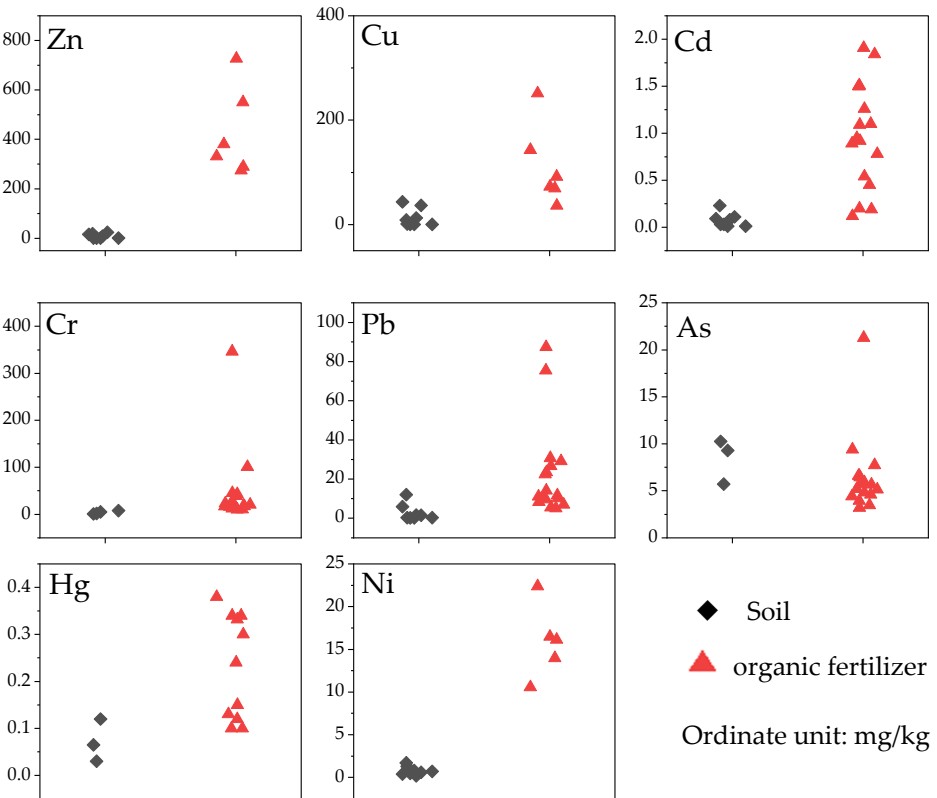

**Figure 3.** The content of eight type heavy metals in soil and organic fertilizer (The abscissa represents medium, soil or organic fertilizer; the ordinate represents each metal concentration.).

According to the definition of heavy metals speciation by Tessier and BCR method, the acid-soluble of the BCR can be regarded as the sum of the exchangeable and carbonate bound of the Tessier. The reducible of the BCR can be regarded as iron–manganese oxide bound of the Tessier, and the oxidized of the BCR can be regarded as organic bound of the Tessier.

For soil, the proportions of all heavy metals in the residue were much higher, with an absolutely high percentage of more than 85% of residue in total, in nearly half the number of the literature; the Fe–Mn oxide-bound of Zn, Cr, Pb and Ni accounted for a relatively high proportion, about 25% of Fe–Mn oxide-bound in total.

The heavy metals speciation in organic fertilizers were mainly residue and organically bound, and the residue was no longer in a dominant position compared to the soil. Only 9% of statistical literature showed the residue had a certain proportion of more than 85% in each total heavy metal, and the organic-bound accounted for about 28.5% in each. It also could be seen that organic matter content in organic fertilizers was much higher than that in soil, and it was undeniable that the organic-bound had a far higher amount of binding point with heavy metals in, so the organic-bound had a larger proportion of heavy metals accordingly.

The speciation of different heavy metals was slightly different in the same environment. As shown in Table 3 the main speciation were the bound to organic with a percentage of 21–69% for Cu, 16–48% for Zn and 7–33% for Cd in organic fertilizers. The ability binding to DOM in different kinds of heavy metals was different, which in turn affected the distribution of organic bindings. Metal cations had a strong affinity with DOM. Cu can speciation strong complexes with organic ligands [78], by contrast, the complexation of Zn with DOM was weaker, while the complexation of Cd with organic matter tends to be weakest [79,80]. For the binding ability of organic and inorganic substances, Zn was more likely to bind to inorganic substances, while Cd and Ni tended to be adsorbed by organic substances [81].

**Table 3.** Main speciation of heavy metals in soil and organic fertilizer.

| Soil | References | Organic Fertilizer | References |
|---|---|---|---|
| Zn | | | |
| Residual (55–87%), Bound to Fe–manganese oxide (8–17%) | [72] | Bound to organic (26–33%), Bound to Fe–manganese oxide (26–35%), Bound to carbonate bonded (26–38%) | [29] |
| Residual (73–88%), Bound to Fe–manganese oxide (9–13%) | [81] | Residual (77%), Oxidizable (16%) | [82] |
| Residual (44-93%), Bound to Fe–manganese oxide (4–42%) | [34] | Exchangeable (33–75%), Oxidizable (15–35%), Reducible (16–33%) | [20] |
| Residual (32%). | [83] | Residual (56%), Bound to Fe–manganese oxide (25%) | [55] |
| | | Residual (31–32%), Reducible (26%), Oxidizable (29–48%) | [84] |
| Cu | | | |
| Residual (97–99%), Exchangeable (1–2.4%) | [72] | Bound to organic (36–44%), Fe–Mn oxide bound (29–32%) | [29] |
| Residual (61–92%), Bound to organic (7–14%) | [81] | Residual (32–69%), Oxidizable (21–54%) | [82] |
| Residual (67–80%), Reducible (14–20%) | [85] | Oxidizable (48–69%), Exchangeable (12–36%) | [20] |
| Residual (35%), Bound to organic (31%) | [83] | Organic bound (67%), Bound to Fe–manganese oxide (16%) | [55] |
| | | Exchangeable (24–31%), Bound to Fe–manganese oxide (26–58%) | [84] |
| Cr | | | |
| Residual (40%), Reducible (22%) | [83] | Residual (58%), Bound to organic (35%) | [55] |
| Residual (46–87%), Bound to Fe–manganese oxide (10–38%) | [34] | Residual (54–77%), Oxidizable (16–29%) | [82] |
| Pb | | | |
| Residual (31-52%), Bound to Fe–manganese oxide (26–57%) | [81] | Residual (35–43%), Oxidizable (30–40%) | [20] |
| Bound to Fe–manganese oxide (33–43%), residual (27–30%) | [69] | Residual (82%), Bound to organic (12%) | [55] |
| Residual (45%), organically bound (30%) | [27] | Residual (91%), Oxidizable (7%) | [82] |
| Organic bound (34%), residual (26%) | [86] | | |
| Residual (54–76%), Fe–Mn oxide bound (10–29%) | [34] | | |

**Table 3.** *Cont.*

| Soil | References | Organic Fertilizer | References |
|---|---|---|---|
| Ni | | | |
| Residual (56–94%), exchangeable (3–30%) | [72] | | |
| Residual (59–88%), Bound to Fe–manganese oxide (10–31%) | [81] | | |
| Residual (87–97%), Reducible (0.8–7%) | [85] | | |
| Residual (36%), Oxidizable (33%) | [83] | Residual (53–63%), Oxidizable (17–32%) | [82] |
| Residual (51–91%), Bound to Fe–manganese oxide (4–22%) | [34] | | |
| Bound to Fe–manganese oxide (39–57%), Exchangeable and bound to carbonate (21–26%) | [84] | | |
| Cd | | | |
| Residual (90–97%), Exchangeable (1.7–7.7%) | [72] | Residual (78%), Bound to organic (7%) | [55] |
| Residual (92–96%), Oxidizable (3–24%) | [85] | Exchangeable (27–71%), Reducible (20–55%) | [20] |
| Residual (35%), Oxidizable (35%) | [83] | Residual (64%), Oxidizable (33%) | [82] |
| As | | | |
| As oxide and As in silicon (55–61%) | [87] | Residual (75–85%), Exchangeable (6–11%) | [55] |
| | | Residual (36–82%), Oxidizable (11–62%) | [82] |
| Hg | | | |
| Residual (46–58%), Bound to organic (18–27%) | [88] | Residual (61%), Bound to organic (21–30%) | [55] |
| Residual (56–63%), Bound to strongly organic (19–22%) | [89] | | |

Metal re-adsorption and redistribution influenced the distribution of heavy metals speciation. The re-distribution was a natural phenomenon of adsorption and desorption equilibrium, in which the heavy metal ions released were re-adsorbed by the particles. Heavy metal ions released into the liquid can be retained by added nitrilotriacetic acid (NTA) to chelate the heavy metal ion, which can effectively avoid the re-adsorption phenomenon [90]. In addition, pretreatment process can also affect the distribution of heavy metals speciation. Fresh preservation, natural air-drying and freeze-drying were three typical pretreatment methods, and they had significant effects on the speciation of Cr, Pb and As in riverine and marine sediments. The speciation, in descending order of affected level, were exchangeable and carbonate-bounds, Fe-Mn oxides, organic-bounds and the residual [91]. Fresh samples tended to cause uneven sample mixing and large weighing errors because of the presence of water, so it was not suitable for test. The freeze drying process was done at low temperature and low pressure. It could exclude the influence of water, microorganisms and dust on the samples, compared with natural air drying. But the requirements of the freeze-drying process were strict, so it was not suitable for the treatment of a large number of samples.

## 6. Conclusions

The extraction procedures of heavy metals speciation in organic fertilizers were classified into two types: single-stage extraction and SEP. There were three main types of heavy metals speciation extraction methods in organic fertilizers: one was the Tessier method and modified Tessier method; the second was the BCR method and modified BCR method; and the third was the other methods. As the different composition between soil and organic fertilizer, the improvements of the modified method of the speciation of heavy metals in

organic fertilizer was taken, included reducing the solid-liquid ratio, changing the type and concentration of extractant, pH, extraction time, centrifugation time, centrifugation speed, solid cleaning procedure, etc. In short, the extractant concentration was reduced and/or the extraction time was extended for the exchangeable state. The extraction time was shortened for the carbonate bound state. The extractant concentration was increased or the extraction temperature was decreased for the Fe/Mn oxide bound state, and the extractant concentration was decreased for the organic bound state. Due to the lack of a complete evaluation system, although BCR method has a certain reference material, which is far from enough, it is difficult to compare the advantages and disadvantages of different methods at present.

According to the result of SEP, the heavy metals speciation in organic fertilizers were mainly residue and organically bound, and the residue was no longer in a dominant position compared to the soil. The change probably stems from the differences in the distribution of heavy metals speciation between soils and organic fertilizers mainly due to the difference in organic matter, minerals and pH value. The pH value was low in soil and fluctuated widely, while pH value was high in organic fertilizer. The acidic environment was prone to the activation of heavy metals, but the effect of pH appears weaker compared to the effect of dissolved organic matter on heavy metals in organic fertilizer. More organic matter will greatly increase the content of heavy metals in the organic bound state, so it may affect the completeness of the organic bound state extraction. Organic matter included humus and non-humic organic matter. The dissolved humus was easy to combine with heavy metal ions to make heavy metal ions easy to extract, thus causing heavy metal activation. But the insoluble heavy metals would make heavy metal ions passivated. Furthermore, the content of heavy metals, especially Cu and Zn, in organic fertilizer was much higher than that in soil. With the characteristics of less binding mineral content and more organic matter content, even some metals with high mineral affinity will increase the content of organic binding state. Therefore, we should pay more attention to the optimization of the morphological analysis method of heavy metals in organic fertilizers.

There are, of course, shortcomings to this review. The information and discussion is limited about the re-adsorption during extraction of heavy metal speciation in different livestock and poultry. Re-adsorption phenomenon is a common problem in sequential extraction, which will affect the determination accuracy. It is related to the properties of materials as well as the content of organic matter. The way to avoid this problem is a necessary topic in future discussion. In addition, different species of livestock and poultry have different composition, and it is not clear whether the classical speciation analysis methods can all be adapted to these materials. Those factors are important in the extraction procedure of heavy metals and need continuous attention in the future.

## 7. Outlook

Information about the extraction accuracy of each speciation among the optimizing method of SEP was scarce in the current study. Most research was focused on whether the concentration of a specific speciation of heavy metal was increasing or not by changing the extraction parameters and did not mention the accuracy of extraction process at all. Some modified methods added a new form of extraction process but ignored the question of whether the extraction of the previous step interfered with the next step. The evaluation criteria of the modified methods are quantitative analysis currently, not qualitative analysis. It may be because there is no standard substance of heavy metal speciation. For the optimization of the BCR method, it can be verified by a recovery rate using a standard substance for BCR extraction process only named GBW07437. But for the modified Tessier method or the other methods, there is no suitable methods in quality control. So, the characterization process of the heavy metal speciation is very important to measure whether the speciation is extracted completely in the extraction process.

Soil has a higher mineral content, so refining the forms of heavy metals bound to minerals is beneficial for understanding the distribution characteristics of heavy metals.

In contrast, organic fertilizer has a lower mineral content and a higher organic matter, thus it will be more conducive to obtain more information of heavy metals speciation by integrating the speciation of heavy metals bound to minerals and refining the speciation of heavy metals bound to organic matter. By referencing the existing methods for extraction of heavy metals speciation and the property of organic fertilizers, some optimized measures are proposed, including that to increase the extraction strength of dissolved organic matter, to weaken the extraction ability of bound to iron and manganese oxides, to avoid the phenomenon of re-adsorption and to adjust the centrifugation time and speed to avoid the sample loss. It also was the optimization directions of extraction procedure of heavy metals in organic fertilizers in future research.

**Author Contributions:** Writing—original draft preparation, X.W.; writing—review and editing, J.W., project administration, Y.S.; validation, Y.S.; supervision, G.L., D.L., J.D., H.C., R.L. and Y.Z. All authors have read and agreed to the published version of the manuscript.

**Funding:** This research was funded by three projects: one is the Project for R&D of Academy of Agricultural Planning and Engineering, Ministry of Agriculture and Rural Affairs, China (Agricultural Planning and Engineering, Ministry of Agriculture and Rural Affairs, China, QX202112), the second one is the Opening Foundation of Key Laboratory of Combination of Planting and Breeding, Ministry of Agriculture and Rural Affairs, China (Heilongjiang Academy of Agricultural Sciences, ZY2020-4), the third one is Beijing Modern Agricultural Industrial Technology System Project (Beijing Innovation Team of the Modern Agricultural Research System, China, BAIC08-2022-FQ03).

**Institutional Review Board Statement:** Not applicable.

**Informed Consent Statement:** Not applicable.

**Data Availability Statement:** Not applicable.

**Conflicts of Interest:** The authors declare no conflict of interest.

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
