# Peer review of "Speciation Analysis Method of Heavy Metals in Organic Fertilizers: A Review"

_sustainability, doi:10.3390/su142416789_

Round 1
Reviewer 1 Report
The manuscript with title "Extraction procedure of heavy metals in organic fertilizers: A review" comprehensive review the extraction nethods, explore the problems of sequence extraction procedure and summarize the factors affecting the distribution of heavy metals speciation. The topic is interesting. The following comments may improve the quality of the manuscript.
1. The keywords is same as the title. Please change them.
2. Please indicated the reasons why you want to review the extraction procedure of heavy metals in organic fertilizers in Introduction section.
3. Line 188-208 please give a reference.
4. Line 302-303 Please put the [57] behind the Zhang et al.
5. Please put the figure 2 in the text. Why do you use the box figure, but not the histogram.
6. Please clearly stated the shortcomings of this research.
Reviewer 2 Report
The manuscript mainly reviews the accumulation of heavy metals in livestock and poultry manure, and the speciation of heavy metals in organic fertilizers and soils. In general, it is informative.
The title of the manuscript is “Extraction procedure of heavy metals in organic fertilizers: A review”. However, I don’t find any discussion involved in the extraction of heavy metals in polluted organic fertilizers.
The accumulation of toxic heavy metals in organic fertilizers commonly occurs. You should find a way to remove heavy metals in polluted organic fertilizers effectively based on many previous studies, rather than emphasize heavy metal speciation.
You cost many words to discuss the speciation of heavy metals in soils and manure. This is an old topic. I don't find any new significant meanings.
The manuscript had better be rewritten compeletely.
Line 38: dependent.
Line 39-40: how to ---????.
Line 284: affecting.
Line 295: [57].
Line 314: [60].
Line 336: Comparison of heavy metal speciation between organic fertilizers and soils.
Line 390-411: the conclusions should be rewritten completely.
Reviewer 3 Report
Dear Authors:
This work aims to comprehensively review the extraction methods. The reviem is well written and each extraction procedure of heavy metals is described in line with the observations. However, there are few changes which are required prior to the acceptance of the manuscript for publication. The suggestions are as follows:
In my opinion, the introduction should be more extensive, explaining a bit more about the problems and also including aspects of emission control in other processes. Thus, it is recommended to talk for example about gasification or combustion of waste, including issues such as emissions. For example: “Control of several emissions during olive pomace thermal degradation”; International Journal of Molecular Sciences, Open AccessVolume 15, Issue 10, 13 October 2014, Pages 18349-18361.
It is also recommended to explain what is going to be developed in the following sections, documenting with bibliography the most interesting aspects of the review.
To be corrected:
Line 37: space must be left between chain and [4].
In the same way, Line 74, space must be left between below 1.5 mg/kg and [10]. Idem, line 165 [31], line 295 [57], line 328 [64]
In Line 111, when it is said that: The "European Community Bureau of Reference Materials", reference should be given.
In line 140, Table 1 should be revised, object: acid. Also, try not to cut the table into two pages already in the editing process.
In my opinio, in addition, in the 4.1 section, references [25-31] should be further explained in the body of the text.
Also, Figure 2 should be referred to in the text as Figure 2. And Fig. 3 in line 337 of the text replace with Figure 3.
Thank you so much.
Best regards.
Reviewer 4 Report
The article is interesting and presents an important amount of information regarding the methods of extraction (removal) of metal ions from organic fertilizers obtained from animal droppings. In my opinion the paper with title: "Extraction procedure of heavy metals in organic fertilizers: A review" could be published in the journal Sustainability if the authors will make the corrections and specifications suggested
The article needs major revision.
1. The abstract is much too short and
presents much too general information. Please develop the Abstract with details regarding the metal ion composition of organic fertilisers, with information related to the methods used to remove metal ions and their performance.
2. Please specify in the text, lines 94-95, what is the meaning of the acronym, DTPA-TEA solution, where it appears for the first time in the text.
3. Please make a clearer structuring of the article so as to highlight what are the characteristics of the type of speciation and what are the methods used for their extraction from fertilizers.
For example in Table 1 Single extraction method for heavy metals. and Table 2 Operation steps of continuous extraction method (SEP) are present in column 1 different type of speciation.
In the text that follows the tables, you give clearer or less clear explanations about what that type of speciation represents and what heavy metals extraction method can be used for heavy metals removal.
Please highlight better in the text what each type of speciation represents and what is the method used to highlight it and to remove metal ions from organic fertilisers. The text is much too compact and does not differentiate between the type of species and the methods applied for the recovery of metal ions.
Possibly introduce sub-chapters to highlight them.
Migration speciation
Migration speciation represents the..... . The methods used for
Effective speciation
Effective speciation represented.......t The methods used for
4. In Figure 3. please specify which parameters vary on the Ox and Oy axes.
5. In chapter 5.1. please highlight better the influence of each factor on the metal ion extraction process.
6. I think it would be useful to introduce some information regarding the efficiency of the methods applied to the removal of metal ions from organic fertilisers.
7 I have not seen any reference regarding the extraction efficiency of metal ions, using different methods depending on the type of ion speciation studied. It would be useful to make a comparison between the extractive efficiencies of the different ions present in organic fertilisers by one method or another.
8. In the Conclusions Chapter, you must present a comparison of the extractive methods according to the reaction medium, according to speciation; conclusions related to the efficiency of the methods of extracting metal ions from fertilizers depending on their speciation and different factors, composition, pH, type of ion, etc.
Round 2
Reviewer 4 Report
In my opinion the paper with title: "Extraction procedure of heavy metals in organic fertilizers: A review" should be published in the journal Sustainability, because the authors made the corrections and specifications suggested.